materials science/optics/nanotechnology

r-GO, nanocomposite, ellipsometry, photoluminescence emission

**Author for correspondence:**
Jai Singh
e-mail: jai.bhu@gmail.com

This article has been edited by the Royal Society of Chemistry, including the commissioning, peer review process and editorial aspects up to the point of acceptance.

# Ultra-bright emission from Sr doped TiO₂ nanoparticles through r-GO conjugation

Sanhita Mandal[1], Neha Jain[1], Mukesh Kumar Pandey[2], S. S. Sreejakumari[4], Prashant Shukla[1], Anupama Chanda[1], Sudipta Som[3], Subrata Das[4] and Jai Singh[1]

[1]Department of Physics, Dr. Harisingh Gour Central University, Sagar, Madhya Pradesh 47003, India
[2]Department of Physics, and [3]Department of Chemical Engineering, National Taiwan University, Taipei 10617, Taiwan, Republic of China
[4]Materials Science and Technology Division, CSIR – National Institute for Interdisciplinary Science and Technology, Thiruvananthapuram, Kerala 695019, India

NJ, 0000-0003-4239-5541; JS, 0000-0001-8293-8921

Graphene and semiconductor nanocomposite garnered much interest in nanoscience and nanotechnology. In this research, TiO₂, TiO₂: Sr and TiO₂: Sr/r-GO (reduced graphene oxide) nanocomposites have been successfully synthesized via a wet chemical synthesis method. The microscopic studies confirmed the formation of graphene sheets which looked like a paper which could easily wrap over the bacterial surface killing them. The optical band gap of these nanocomposites is determined by UV–visible absorption spectra which inferred that optical band gap decreases with Sr²⁺ incorporation and r-GO attachment. Furthermore, photoluminescence (PL) study revealed that the intensity of emission is prominent for TiO₂: Sr/r-GO. The enhancement in PL intensity with r-GO is due to creation of more oxygen vacancies and defects which generally capture the photoinduced carriers inhibiting recombination rate of free carriers promoting the photocatalytic reactions.

## 1. Introduction

TiO₂ nanoparticles have several unique features such as low cost, abundance, chemical stability, wide band gap, non-toxicity, environment friendly and large exciton binding energy (approx. 60 meV) [1–5]. The optical and biological performances of TiO₂ are also quite good. These are also useful in dye sensitized solar cells (DSSCs), photocatalysis, gas sensor, lithium batteries and biosensors [6]. TiO₂ photocatalyst has vast applications in air

and water purification as well as in deodorization, sterilization and soil proof [7,8]. For good photocatalytic activity, the recombination of photo-generated electron hole pairs should be very slow. However, for $TiO_2$ photocatalyst recombination rate is fast which limits its applicability. Again, band gap of $TiO_2$ falls in UV-range and for a good photocatalyst, its absorption should be in visible region. Various efforts have been made to increase its absorption in visible region as well as to reduce its band gap. When any other metal is doped in $TiO_2$, oxygen vacancies are created as well band gap of $TiO_2$ getting reduced, which affect optical performance of $TiO_2$. The variation in band gap with doping of transition metals was reported by several researchers. The dopants form an intermediate energy level so that band gap gets decreased. The doping of $Sr^{2+}$ improves magnetic and optical properties. It also creates oxygen vacancy in crystal which could affect optical properties. Although, emission losses occur from surface of $TiO_2$: $Sr^{2+}$ due to presence of non-radiative decay centres, it can be resolved by protecting surface of $TiO_2$: $Sr^{2+}$ by graphene or any other shell.

Reduced graphene oxide has a two-dimensional crystal structure with a single atom thickness and is one of the most promising materials in the field of nanoscience and technology [9]. If reduced graphene oxide can be attached on the sample surface then non-radiative defects could be removed from the sample so that its optical performance gets improved. It was first prepared in 2004 by peeling a single layer of graphite using sticky tape and a pencil [10]. It is an important material in the field of nano-technology because of its structure and very large surface area [11]. Also, its electron conductivity is proved to be the ideal material for synthesis of nanocomposites for improving antibacterial properties [12–15]. Absorption of single stranded DNA onto graphene sheets to quench electron donors, the ability of graphene to prevent the biomolecules from enzymatic cleavage, as well as transportation facility in living cells and *in vivo* systems, have revealed the potential of graphene application in biological studies and biotechnology [16–18].

Herein, $TiO_2$: Sr/r-GO nanocomposites were synthesized to investigate their optical and antibacterial properties. The XRD analysis illustrated the crystallinity and successful substitution of $Ti^{4+}$ with $Sr^{2+}$ ions. The various vibrational modes of Ti-O and graphene were examined by FTIR. The formation of nanocomposite was also examined via HRTEM and EDX. Comparative optical studies were also carried out on $TiO_2$, $TiO_2$: Sr and $TiO_2$: Sr/r-GO samples via measuring their band gap energy, photoluminescence, chromaticity etc.

# 2. Experimental details

## 2.1. Materials

Flake graphite powder, $H_2SO_4$ (98 wt%, Merck), $NaNO_3$ (99.9%, Merck), NaOH (Rankem), dilute HCl (65%, Merck), $H_2O_2$ (30%, Merck), $KMnO_4$ (99.8%, Merck), titanium di-isopropoxide (75%, Merck), PVP (Alfa Aesar), strontium nitrate ($Sr(NO_3)_2$, 99.98%, Merck) and sodium borohydride (Merck).

## 2.2. Synthesis of reduced graphene oxide

The mixture of flake graphite powder and $NaNO_3$ was prepared in weight ratio of 2:1, respectively. The mixture was added into beaker with 10 ml of 98 wt% $H_2SO_4$ at 15°C and a suspension was obtained. Then $KMnO_4$ which acted as oxidizing agent was gradually added into the suspension with continuous stirring. The weight of $KMnO_4$ powder is three times as much as one of the mixtures. There were three steps for the following process. First of all, it is low temperature reaction. The temperature of the reaction was controlled below 20°C for 2 h; at the same time the suspension should be stirred continuously. The second step is the mid temperature process. The temperature of the mixture was maintained at 35°C for 30 min after $KMnO_4$ was totally dissolved. Finally, it is high temperature reaction. A certain amount of deionized water was added into the mixture slowly and therefore a large amount of heat was released when concentrated $H_2SO_4$ was diluted; 15 min later, 200 ml of water followed by 30% $H_2O_2$ was added into the mixture with continuous stirring. The dark greenish coloured suspension was filtered by qualitative filter paper when it was still hot and the solid mixture was washed with dilute HCl aqueous and distilled water and dried in vacuum at 70°C for 6 h.

## 2.3. Synthesis of TiO$_2$: Sr, TiO$_2$: Sr/GO

For the $TiO_2$ nanoparticle synthesis, 5 ml of titanium di-isopropoxide was dissolved in 10 ml of deionized water to obtain a solution of 0.2 M concentration. The solution was stirred continuously using magnetic

stirrer until a homogeneous solution was obtained. Then 1 g of polyvinylpyrrolidine (PVP, MW 40 000) was added into the titanium di-isopropoxide solution, as a capping agent. Finally, 2.8 g of NaOH was dissolved in 7.5 ml of deionized water (0.8 M) and this solution was slowly added into PVP modified titanium precursor solution. Stirring was continued for 2 h and the white precipitate thus obtained was rinsed with deionized water several times and filtered. The resultant product was dried at 60°C for 4 h. For the $TiO_2$: Sr nanoparticle synthesis, 0.3834 g of strontium nitrate $(Sr(NO_3)_2)$ was added to keep 5 wt% (0.01 M) of Sr in the starting solution and the same procedure was followed as in the case of bare $TiO_2$.

For $TiO_2$: Sr/r-GO sample synthesis, 30 mg of graphene oxide was dispersed in 50 ml of deionized water and sonicated for 1 h. Then PVP modified titanium precursor solution with 5 wt % of Sr was added into the GO solution under magnetic stirring followed by addition of 4 ml of 0.0008 M $NaBH_4$ solution. Here $NaBH_4$ acts as a reducing agent. The remaining synthesis procedures were the same as that of the bare $TiO_2$ synthesis. All the products were calcined at 400°C for 3 h.

## 2.4. Characterizations

The crystal structure of the synthesized samples was studied by using D8 Bruker X-ray diffractometer (XRD) with Ni-filtered Cu-K$\alpha$ (1.5405 Å) radiation at 40 kV and 40 mA. Fourier transform infrared (FTIR) spectra were observed using spectrophotometer (Bruker, Alpha T, Germany). The microstructure has been examined by transmission electron microscope (TECNAI G2). The adsorption spectra were measured by the SYNTRONICS double beam UV–Vis spectrophotometer: 2201 (bandwidth = 3.0 nm). Photoluminescence spectra were obtained by using spectro-fluorometer (VARIAN-CARY Eclipse). Ellipsometry measurement in the wavelength range of 400–800 nm in steps of 2 nm was performed by J. A. Woollam V-VASE ellipsometer spectroscopy at room temperature.

# 3. Results and discussions

## 3.1. X-Ray diffraction technique

The as-obtained $TiO_2$ nanoparticles exhibit a mixture of three phases such as anatase, brookite and rutile with the majority of anatase phase, as shown in figure 1a [19]. However, with the addition of Sr and further by r-GO, the percentage of brookite and rutile phase are found to be increased. Meanwhile, after incorporating Sr to $TiO_2$ host, small peaks due to the presence of $SrCO_3$ are observed as seen in figure 1b,c. The formation of $SrCO_3$ may be due to the decomposition of $SrNO_3$ into SrO on the surface of $TiO_2$, which later converted into $SrCO_3$ after reacting with $CO_2$ during the annealing at 400°C. The as-formed polycrystalline $SrCO_3$ might be responsible for the formation of brookite phase (figure 1b).

The incorporation of graphene broadened the XRD peaks and facilitates the rutile phase owing to the reduction in crystalline size, as shown in figure 1c. It is reported that the anatase to rutile phase transformation is an extremely size dependent effect and graphene usually prevents the growth of grains/crystallites. The particle agglomerations get prevented on the surface of graphene. Reportedly, during the formation of graphene nanocomposite, graphene is formed and acts as the thin base material for the other coexisting components and keeps them in dispersed form [20]. Meanwhile, the XRD of r-GO is also plotted in figure 1d, which shows two characteristic diffraction peaks at 29.6° and 42.6° attributed to the standard (002) and (100) planes of r-GO [21]. Comparing the XRD of r-GO and the XRD of the $TiO_2$: Sr/r-GO nanocomposite, it can be seen that no diffraction peaks of r-GO are observed in the XRD pattern of the nanocomposite, which might be due to the small amount r-GO in $TiO_2$: Sr/r-GO. Furthermore, the replacement of a lower sized $Ti^{4+}$ ion via a larger sized $Sr^{2+}$ generates volume compensating oxygen vacancies, which altered the atomic distances that led to distortion of the structure and favoured the brookite structure to be formed [22]. It is difficult to substitute smaller radius $Ti^{4+}$ (0.75 Å) via a larger radius $Sr^{2+}$ ion (1.18 Å). Furthermore, the substitution of $Ti^{4+}$ on $Sr^{2+}$ could generate two oxygen vacancies for charge neutrality in the lattice. Considering the larger size of $Sr^{2+}$ ions, it can be predicted that besides $Sr^{2+}$ at the $Ti^{4+}$ sites, a large fraction of ions might stay on the surface or move to the interstitial position.

It is also very tough to understand whether $Sr^{2+}$ successfully replaced $Ti^{4+}$ ions or not. However, a shift of the main XRD peak at $2\theta \sim 25.22°$ in the diffraction pattern of $TiO_2$ to lower $2\theta$ values with the incorporation of $Sr^{2+}$ ion has been observed (as depicted in figure 2), which may indicate that some of $Sr^{2+}$ ions are successfully incorporated into $Ti^{4+}$ sites.

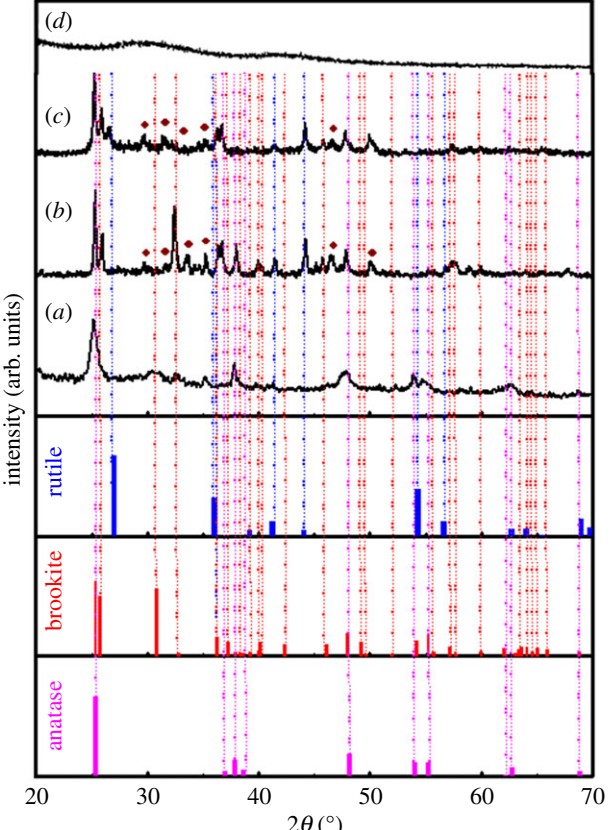

**Figure 1.** XRD patterns of (*a*) TiO$_2$, (*b*) TiO$_2$: Sr, (*c*) TiO$_2$: Sr/r-GO, (*d*) r-GO.

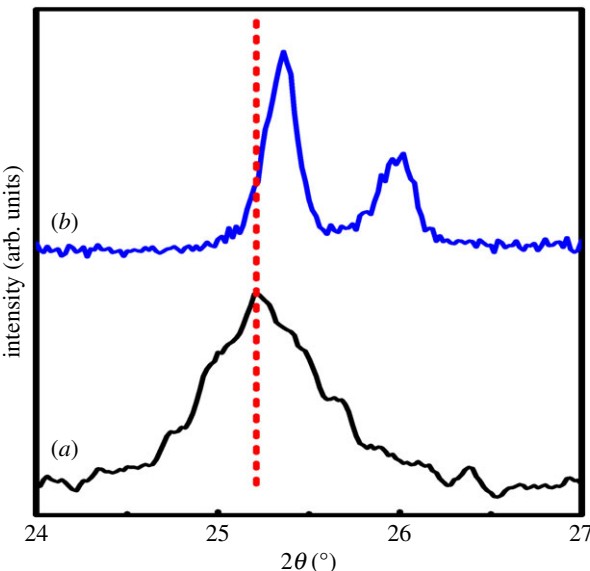

**Figure 2.** Magnifying XRD patterns of (*a*) TiO$_2$, (*b*) TiO$_2$: Sr in the region between 24° and 27°.

A good number of researches directed the formation of mixed phase TiO$_2$ (anatase and rutile) at low temperatures below 450°C has been reported so far. For example, Wang *et al.* produced rutile nanorods via the direct hydrolysis of TiCl$_4$ ethanolic solution in water at as low as 50°C [23]. Fischer and co-workers produced a mixture of TiO$_2$ nanoparticles via low-temperature (130°C) dissolution-precipitation on a microfiltration PES membrane with mixed phase compositions of anatase, brookite, and rutile [24]. Lijuan Bu *et al.* synthesized rutile TiO$_2$ nanoparticles via treating anatase TiO$_2$ with concentrated HNO$_3$ under the hydrothermal conditions operated at 180°C for 24 h [25]. According to the above discussion, it

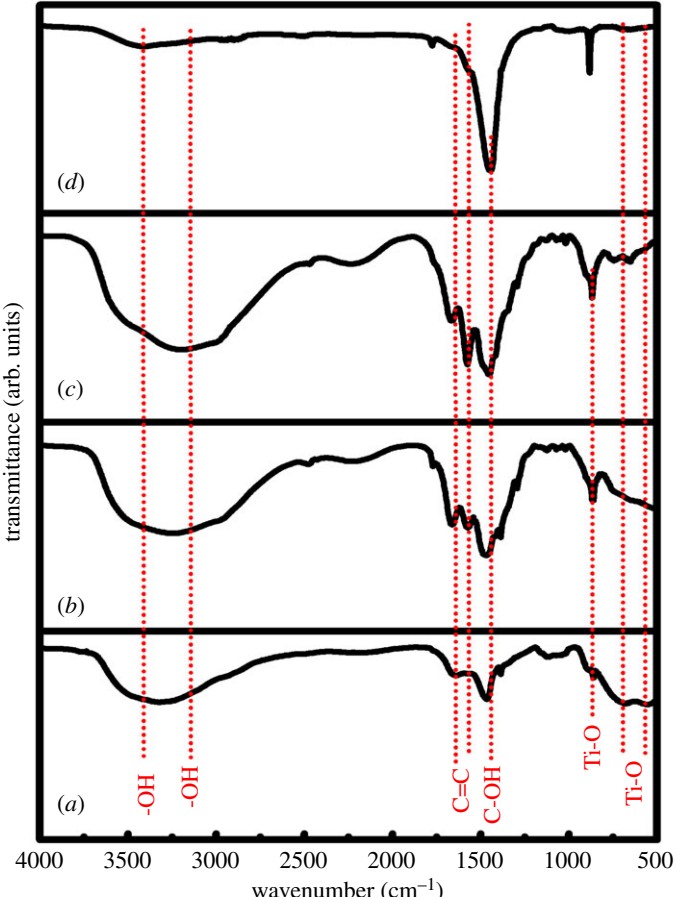

**Figure 3.** FTIR spectra of (*a*) r-GO, (*b*) TiO$_2$, (*c*) TiO$_2$: Sr, and (*d*) TiO$_2$: Sr/r-GO.

can be concluded that the formation and the transformation of rutile phase is mainly dependent upon the precursor materials and synthesis conditions. It can be seen from figure 1 that the diffraction peak intensities are enhanced with the incorporation of Sr and r-GO into TiO$_2$, which indicates that the crystallinity as well as the crystalline size also increased. The crystallite sizes of the present samples are also calculated using Scherer formula: $D = 0.89\lambda/(\beta \cos \theta)$, where $\lambda$ is the X-ray wavelength, $\beta$ is the half width of the main XRD peak, and $\theta$ is the Bragg angle [26]. The crystallite sizes ($D$) are calculated to be around 20.31, 42.96 and 44.42 nm for TiO$_2$, TiO$_2$: Sr, and TiO$_2$: Sr/r-GO, respectively.

## 3.2. Infrared studies

Figure 3 shows FTIR spectra of the present samples including the r-GO. As seen from figure 3*a–d*, the transmittance peaks at around 551 and 682 cm$^{-1}$ are due to the Ti-O stretching vibrations of TiO$_2$ whereas the peak at 880 cm$^{-1}$ is corresponding to TiO$_6$ octahedron bending vibration [27,28]. The absorbances in range 1660–1680 cm$^{-1}$ are assigned to the C=C bond, as shown in figure 3 [29], while the absorbance at 1475 cm$^{-1}$ can be attributed to the C-OH vibration. The intensity of these two vibrational bands is higher in TiO$_2$: Sr/r-GO nanocomposites. Such bands are also seen in TiO$_2$ and TiO$_2$: Sr, which might have attributed to the residual carbon content introduced from some precursors such as polyvinylpyrrolidine. Moreover, the intensity and the position of the vibrational peaks between 1000 and 500 cm$^{-1}$ enhanced and shifted slightly to the lower wavenumber in the case of TiO$_2$: Sr/r-GO nanocomposites, which might be due to the interaction between r-GO and TiO$_2$ nanoparticles [30]. The broad spectrum at 3314 cm$^{-1}$ is assigned to the O-H stretching vibrations of the H$_2$O molecules [27].

## 3.3. Morphological studies

The TEM image of TiO$_2$ nanoparticles represented in figure 4*a* and that of the composite represented in figure 4*b* revealed that the size of TiO$_2$ nanoparticles is not uniform. Their sizes vary between several

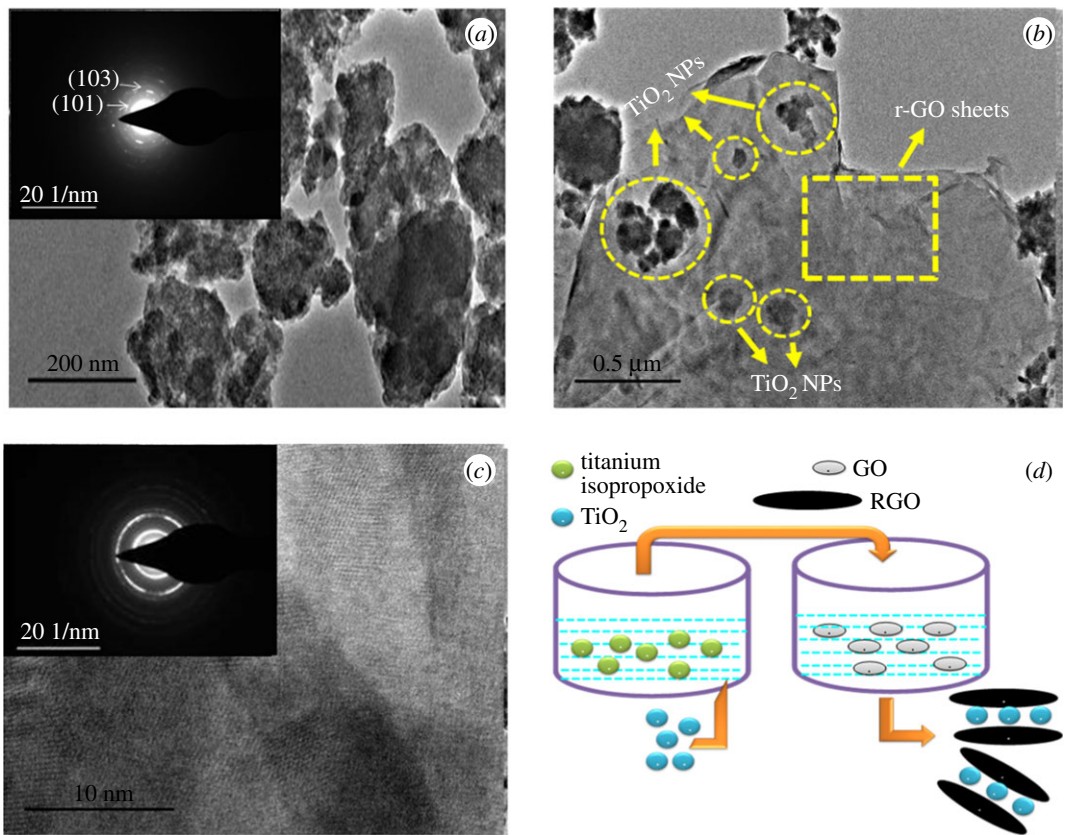

**Figure 4.** (*a*) TEM image and SAED pattern of TiO$_2$ nanoparticle, (*b*) TEM image of TiO$_2$: Sr/r-GO. (*c*) HR-TEM and SAED image of graphene. (*d*) Synthesis mechanisms of TiO$_2$:Sr/r-GO.

nanometres ranging from 40 to 120 nm. From the SAED patterns of the TiO$_2$ nanoparticles, lattice planes (101) and (103) of the anatase phase could be identified, as shown in the inset of figure 4*a*. Moreover, the morphology of the nanoparticles is irregular with rough edges. Meanwhile, the HR-TEM image of the r-GO elaborates multilayer sheet, as shown in figure 4*c*, indicating that an effective exfoliation of graphene has taken place. TEM image of r-GO also shows curved sheet-like morphology with smooth surface. The inset of figure 4*c* represents the SAED pattern of r-GO. As seen from this SAED pattern, r-GO exhibited a spot pattern rather than a ring pattern indicating that the sample consists of r-GO sheets with crystalline nature. This structural nature of r-GO might be a reason for their effective slicing of the bacterial cells.

A reported result indicates that for ZnO the antibacterial property is size dependent [31] and for TiO$_2$ also the biocidal activity can be attributed to the size of the nanoparticles. With the addition of Sr$^{2+}$ the size of the nanoparticles was reduced. The TEM image shown in figure 4 inferred the paper-like graphene sheets over which the TiO$_2$ nanoparticles are oriented. The edges of the graphene sheet are blade-like structure which could easily slice the bacterial cell membrane leading to death. r-GO having sheet-like structure can wrap over the bacteria easily [16]. The combined effect of graphene and TiO$_2$ showed synergetic effect in enhancing the biocidal property. Wang *et al.* have shown that the energy barrier for three-layer graphene sheets with corner sites to pierce through the lipid bilayer is greater than the monolayer sheets of same lateral size [17]. Akhavan *et al.* in 2010 discovered that direct contact between bacteria and extremely sharp edge of graphene nanosheets could result in loss of bacterial membrane integrity and leakage of RNA [16,32]. TEM result shows the closer view of the atomic planes of graphene.

## 3.4. UV–visible absorption analysis

Figure 5 shows the UV–Vis diffuse reflectance spectra of the prepared samples. As seen from figure 5, the absorbance of all the samples decreased in the range of 330–400 nm, and the absorption peak around 266 nm has been attributed due to the exciton absorption band [33–37]. With the doping of strontium and reduced graphene oxide, there was not much alteration in the absorption peak. Absorption limit of

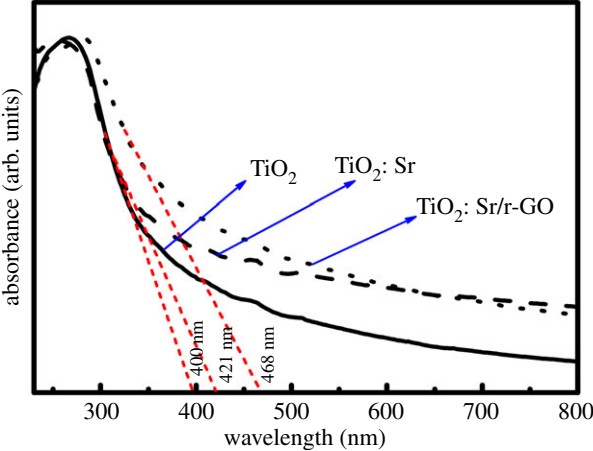

**Figure 5.** UV–Vis diffuse reflectance spectra of TiO$_2$, TiO$_2$: Sr and TiO$_2$: Sr/r-GO.

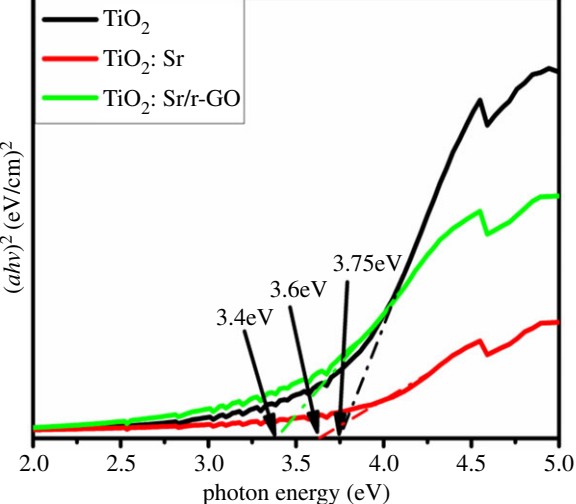

**Figure 6.** Band gap measurement by $(\alpha h\nu)^2$ Vs photon energy ($h\nu$) plot of as synthesized TiO$_2$, TiO$_2$: Sr, TiO$_2$: Sr/r-GO.

TiO$_2$: Sr and TiO$_2$: Sr/r-GO are observed to be 421 and 468 nm, which are higher than that of the undoped sample (400 nm), as observed in figure 5. Such results indicate that the TiO$_2$: Sr and TiO$_2$: Sr/r-GO have smaller band gap energy than that of the undoped sample. Overall, the UV–visible diffuse reflectance spectroscopy characterization suggests that a narrower band gap was achieved by doping TiO$_2$ with strontium as designed. The band gap value is estimated by using the Wood and Tauc equation which is given by $\alpha h\nu = K(h\nu - E_g)^n$, where $\alpha$ is the absorption coefficient, $\nu$ is the frequency of absorbed photon, $h$ is Planck's constant, and $E_g$ is the optical band gap. Exponent $n$ value is chosen for type of optical transition, i.e. $n = \frac{1}{2}$ and 2 are chosen for direct and indirect allowed transitions respectively. In our case the optical band gap was calculated for direct allowed transition ($n = 1/2$) of TiO$_2$. From the respective absorption spectra of each samples, the band gap values were calculated by taking $(\alpha h\nu)^2$ and $h\nu$ plot as shown in figure 6. As can be seen from figure 6, with addition of strontium there occurred a decrease in the band gap from 3.75 to 3.6 eV, which further decreased to 3.4 eV with the r-GO addition. The r-GO addition contributed to the presence of free graphitic carbon which leads to an increase in absorbance. This can be explained as the increase in surface electric charge of oxides in the composite leading to the changes in the formation of electron hole pair formation due to irradiation. This property of increased absorption in the visible region resulted in the increased photocatalytic activity of the sample.

## 3.5. Photoluminescence studies

The photoluminescence emission results of the synthesized samples revealed the charge separation efficiency of the prepared materials. All samples showed strong absorbance around 266 nm, which has

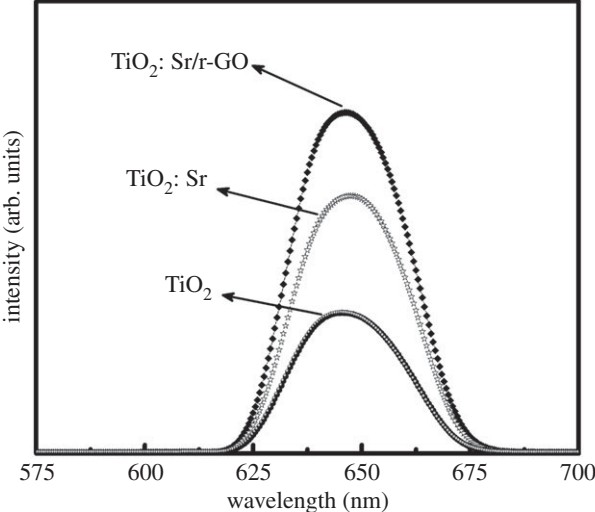

**Figure 7.** Photoluminescence of TiO$_2$, TiO$_2$: Sr, TiO$_2$: Sr/r-GO.

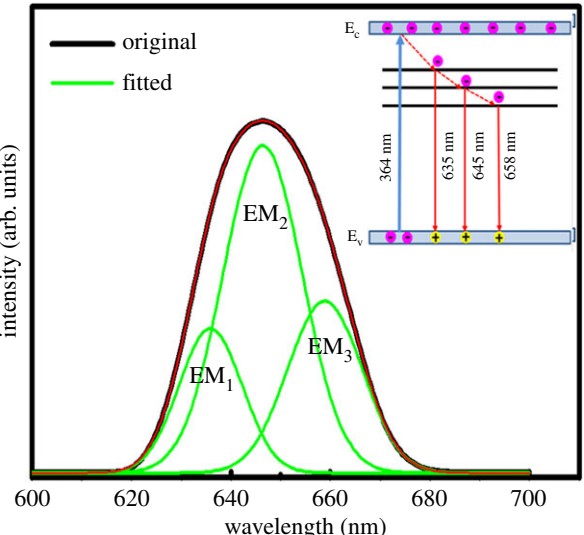

**Figure 8.** Deconvoluted PL emission curve of comparison of TiO$_2$: Sr/r-GO nanocomposite. Inset: schematic energy level diagram of TiO$_2$ for the illustration of the emission mechanisms.

been attributed to the exciton absorption band. Figure 7 shows the PL spectra of all samples taken from 575 to 700 nm. It can be seen that all the samples exhibit intense and broad emission band peaked at around 647 nm, and the intensity increased significantly with the addition of strontium and r-GO. The high surface to volume ratio of TiO$_2$ nanoparticles results in tremendous influence of surface defects and contacting media on their performance in photocatalysis and solar energy conversion. These defects result in deep I trap gap states that impede carrier transport; these are the bad traps. But shallow traps may contribute to the carrier transport, which occur via the rather inefficient process of diffusion so these may be considered the good traps. In order to collect more insight about the nature of such defect states, the PL spectrum of the TiO$_2$: Sr/r-GO nanocomposite has been deconvoluted with multi-peak Gaussian fitting method, as shown in figure 8. As seen from figure 8, the PL spectrum of TiO$_2$: Sr/r-GO nanocomposite can be fitted with three Gaussian sub-bands peaked at 635 nm (EM$_1$), 645 nm (EM$_2$) and 658 nm (EM$_3$). The emission sub-band peaked at 635 nm can be attributed to the self-trapped excitons localized on TiO$_6$ octahedra, and the PL sub-bands at 645 and 658 nm might have originated from the oxygen vacancies. As direct band gap semiconductor, and illumination with ultraviolet light results in broad visible photoluminescence (PL) arising from oxygen vacancies.

The normal emission of anatase particles are dominated by PL arising from recombination of trapped electrons from the valence band holes, leading to a broad spectrum with a peak in the red. The PL

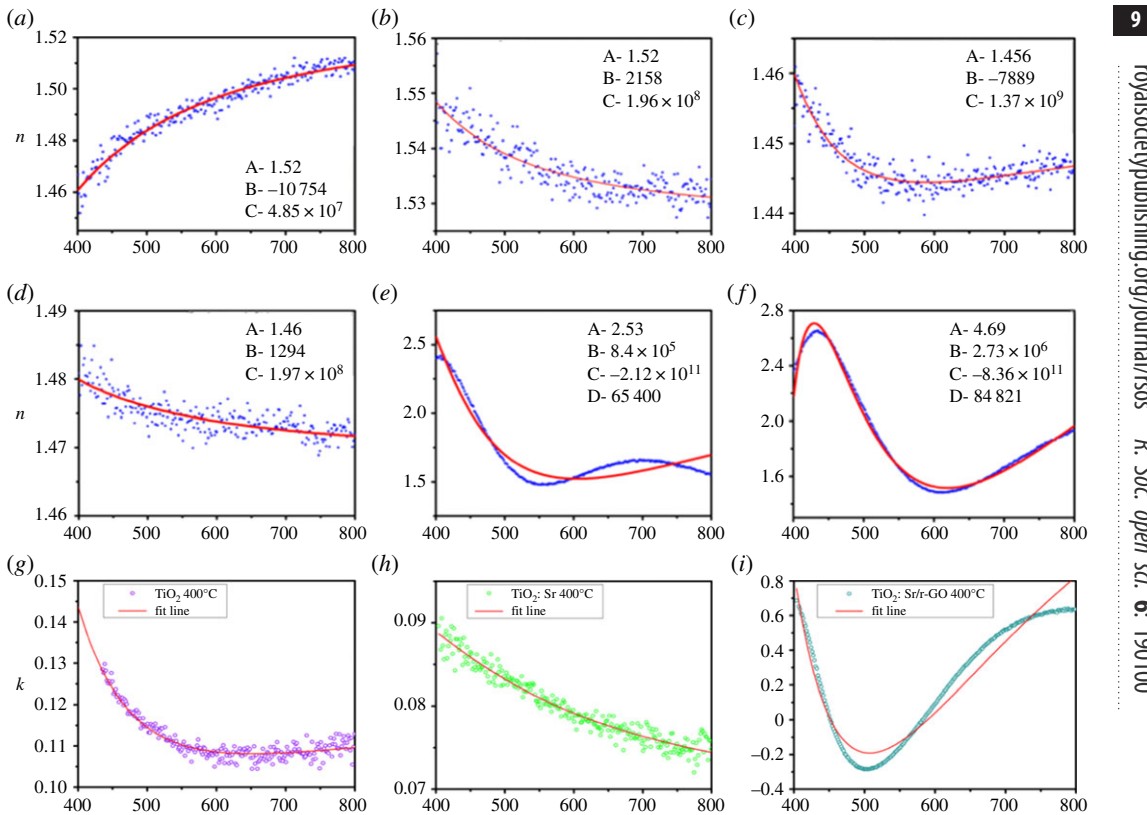

**Figure 9.** Refractive index of TiO$_2$: Sr/r-GO for as-synthesized and annealed samples in the form of thin film (*a*) TiO$_2$, (*b*) TiO$_2$ 400°C, (*c*) TiO$_2$: Sr, (*d*) TiO$_2$: Sr 400°C, (*e*) TiO$_2$:Sr/r-GO and (*f*) TiO$_2$: Sr/r-GO 400°C (*g*−*i*) wavevector versus wavelength plot for TiO$_2$: Sr/r-GO.

emission is directly related to the recombination of excited electrons and holes, owing to which it can be predicted that the higher PL emission intensity indicates more rapid recombination rate which is not advantageous for the high photocatalytic activity. The PL intensities of TiO$_2$: Sr/r-GO and TiO$_2$: Sr are appreciably higher than that in TiO$_2$. This result is very similar with a reported result where the enhanced intensity is reported for TiO$_2$/r-GO [38]. According to this report, TiO$_2$: Sr/r-GO and TiO$_2$: Sr can produce more photo-generated electron-hole pairs, which are responsible for the enhanced PL intensity in TiO$_2$: Sr/r-GO and in TiO$_2$: Sr than that in TiO$_2$. During photoluminescence, photo-induced electrons are prohibited from forming free or binding excitons owing to the binding via the oxygen vacancies and defects. Because of this, the PL emission occurs. Therefore, with the increase in oxygen vacancies or defects, the PL emission intensity also enhances usually. The oxygen vacancies and defects normally capture photo-induced electrons inhibiting the recombination of such photo-induced electrons and holes, which is advantageous for photocatalytic reactions. Simultaneously, oxygen vacancies are advantageous in promoting the oxygen adsorption. Such adsorbed oxygen later interacts strongly with the photo-induced electrons bound by oxygen vacancies. Hence, oxygen vacancies and defects usually favour the photocatalytic reactions in which oxygen dynamically promotes the oxidation of organic materials [39–41]. Overall, the defect centres and vacancies responsible for the intense and broad PL emission from the present samples can be understood from the schematic energy level diagram presented in the inset of figure 8. As seen from this figure, the photo-generated electrons are initially excited to the conduction band of TiO$_2$ via the UV excitation at 300 nm and then relaxed to the defect states.

## 3.6. Ellipsometry analysis

Spectroscopic ellipsometry was used to study the refractive index of the films [42–44]. Figure 9*a*−*f* shows the variation in the refractive index (*n*) with wavelength in the visible region for undoped, Sr-doped and r-GO/Sr co-doped TiO$_2$ films. In this ellipsometric data it can be observed that there is a significant change in the refractive index for a thin film prepared by mixture of TiO$_2$, Sr and r-GO from undoped

$TiO_2$ and Sr-doped $TiO_2$. The overall value of refractive index increases for r-GO-doped film over entire range of wavelength in the visible region. In this range, refractive index value lies in the range of 1.5–2.6 [45]. The enhancement in the index values at higher side for r-GO/Sr co-doped $TiO_2$ film may have arisen due to the change in packing density of the films as well as increase in the carrier concentration. Further, from figure 9b,d,f we can observe that on annealing the samples, the refractive index increases for undoped as well as for doped samples relative to unannealed thin films. The experimental curves are fitted by using the following equations:

$$n = A + \frac{B}{\lambda^2} + \frac{C}{\lambda^4} \tag{3.1}$$

$$\text{and} \quad n = A + \frac{(B-D)}{(D-\lambda^2)} + \frac{(C-D)}{(D-\lambda^4)}, \tag{3.2}$$

where, $A$, $B$, $C$ and $D$ are constants, $\lambda$ is wavelength and $n$ is refractive index. Figure 9a−d is fitted by using equation (3.1) while figure 9e,f is fitted by using equation (3.2). The constant values obtained from fitting are mentioned in respective figures. Increase in refractive index reflects that annealing causes more crystallization of samples.

# 4. Conclusion

The $TiO_2$ nanocomposite can be effective in showing effective antibacterial property. The comparative study between the bare $TiO_2$, $TiO_2$: Sr and $TiO_2$: Sr/r-GO showed that with the subsequent addition of Sr and graphene to the system there occurred morphological changes which resulted in the hike of photocatalytic property. Also the addition of Sr and r-GO helped in reducing the band gap energy of the samples. The substitution of Ti by Sr resulted in oxygen vacancies which was further enhanced with the addition of graphene. The oxygen vacancies and defects normally capture photo-induced electrons inhibiting the recombination of photo-induced electrons and holes which is advantageous for photocatalytic reactions. On investigating all of its major properties it could be concluded that the nanocomposite could be the ideal member in using the green technology for cleaning environment.

Ethics. No special permission was required to carry out this work as it does not include any animal or plant.

Data accessibility. All experimental data are reported in figures 1–9. It has no additional data.

Authors' contribution. S.M. synthesized all samples and collected experimental data for analysis. N.J. and P.S. interpreted ellipsometry data and analysis. S.S.S. and M.K.P. helped to analyse structural data and morphology of the sample. S.D. and S.S. completed writing work of manuscript. J.S. and A.C. revised the article and proposed many good suggestions.

Competing interests. We have no competing interest.

Funding. We have no funding to support this project.

Acknowledgements. The authors cordially thank the Sophisticated Instrument Laboratory of the University for providing various characterization facilities. The authors are also grateful to the CSIR, IMMT, Bhubaneswar for giving the chance to use FTIR and photoluminescence studies.

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
