## [Reviewer comments · Royal Society Open Science]

Review History

RSOS-181430.R0 (Original submission)

Review form: Reviewer 1

Is the manuscript scientifically sound in its present form?

Yes

Are the interpretations and conclusions justified by the results?

Yes

Is the language acceptable?

Yes

Is it clear how to access all supporting data?

Yes

Do you have any ethical concerns with this paper?

No

Have you any concerns about statistical analyses in this paper?

No

Recommendation?

Accept with minor revision (please list in comments)

Comments to the Author(s)

Mondal et al have investigated the comparative study of TiO₂, TiO₂: Sr and TiO₂: Sr/r-GO (reduced graphene oxide) nano-composites for their optical and structural properties for photocatalytic and antibacterial applications. The study is interesting and can be accepted in this journal after a careful revision: The comments are as below:

1. The authors have discussed the possible applications in photocatalytic and antibacterial applications but in the introduction the appropriate references are missing (<https://doi.org/10.1116/1.4904503>, <https://doi.org/10.1116/1.4904729>). TiO₂ is being used in various applications so include one paragraph by highlighting the same.
2. Figure 1 is very busy and the vertical lines are not clear so please replace the graph with higher resolution. Figure 2 has also same issue.
3. Author should estimate crystalline size of TiO₂:Sr and its nanocomposite with r-GO. The crystalline size compare with TEM particles size.
4. FTIR spectra of nanocomposite given in figure 2 shows variation in transmittance for TiO₂, TiO₂:Sr, TiO₂:Sr/r-GO. Author should explain variation in intensity of Transition and include shift observed in wavenumber for nanocomposites.
5. In Ellipsometry data, I would suggest author should fit it with suitable equation to explain variation in refractive index (n) and wave vector (k) with wavelength.
6. Author should add some recent references to compare their finding with references.

Review form: Reviewer 2

Is the manuscript scientifically sound in its present form?

No

Are the interpretations and conclusions justified by the results?

No

Is the language acceptable?

Yes

Is it clear how to access all supporting data?

No

Do you have any ethical concerns with this paper?

No

Have you any concerns about statistical analyses in this paper?

I do not feel qualified to assess the statistics

Recommendation?

Reject

Comments to the Author(s)

The authors of the manuscript "Ultra bright emission from Sr doped TiO₂ nanoparticles through r-GO conjugation" have discussed the enhancement of defect-related PL emission after doping and rGO conjugation. I do not recommend the acceptance of the manuscript based on the following points.

1. The authors have prepared TiO₂ at a calcination temperature of 400 °C. They got a mixture of three peaks; anatase, rutile and brookite. It is understood that anatase and brookite are the metastable phases and can be formed at low calcination temperature and in a low reaction temperature too. But, the formation of rutile phase needs to be supported with proper evidence. Rutile being the most stable form of TiO₂ are usually formed at high calcination temperature.
2. The line "Sr facilitate the brookite phase might be due to polycrystalline nature of Sr related compounds". It is not clearly presented. What types of compounds? SrO, SrCO₃ or what? Similarly, graphene helps in the rutile phase formation. Similarly, Graphene helps in the rutile phase formation. The reason provided as due to "...rutile phase owing to the reduction in crystallite size". How is it so?
3. The authors have mentioned that the location of Sr²⁺ is not clear. This can be understood since one may need several other techniques to verify this. However, the authors can provide some more insights, such as they should mention the ionic size of the dopant and host lattice ions. TiO₂ has TiO₆ as the basic unit. Sr²⁺ has a sufficiently large ionic radii in 6 coordination which is nearly 132 pm. Ti⁴⁺ in 6 coordination has radii of approx. 74 pm. It is quite evident that it would be hard for Sr²⁺ to replace Ti⁴⁺. Thus, instead of Sr²⁺ on Ti⁴⁺ it might be on the interstitial site also. Furthermore, a difference in the charge on Sr and Ti might also create oxygen vacancies in the system for charge balance. The authors need to consider these points because finally they have related PL with oxygen defects.
4. There is a difference between band-to-band absorption and band gap absorption? The authors have claimed 300 nm is the band gap absorption of TiO₂ which correspond to 4.13 eV. However, the band gap as they measured from the Tauc plot gives a band gap of 3.75 eV. Is it the direct or indirect band gap they are calculating?
5. The authors have drawn a line on the absorption edge on the X-axis with Y=0, and mentioned that to be absorption limit? What is the meaning of this? Usually, the line drawn on the absorption edge on the X-axis with Y=0 gives an idea of the band gap of the samples. Are they trying to mean this or what? If that is the case a matching between Fig. 4 and Fig. 5 might happen if the $(ah\nu)^{1/2}$ is considered. The authors have to be clear on this.
6. They have mentioned that the absorption at 300 nm is due to recombination of carriers. What I understand, the photoexcited carriers recombine radiatively or nonradiatively giving light or heat. Now, the recombination process should give emission. Although they have mentioned that in the last line of the PL study that PL is obtained by UV excitation at 300 nm, They have mentioned that recombination is the result for the absorption at 300 nm. How is it so???
7. They have mentioned that the indirect band gap of TiO₂ does not allow a bandgap emission. Thus, the authors need to be more clear now to support that the band gap they have calculated in Fig. 5 is direct or indirect.
8. The authors have mentioned that after Sr addition the PL intensity of TiO₂ is intensified. They surmised that a higher recombination rate is beneficial for high photocatalytic activity. Now, photocatalytic activity requires available electron and holes. If the recombination rate is faster, the electrons and holes will not be available for participation in photocatalysis leading to poor photocatalytic activity. They should be more clear on this line?
9. Why is the enhancement in PL after rGO addition? Is it because of the fluorescence contribution from rGO or due to enhancement of oxygen vacancies?

NOTE: The figures are poorly formatted which makes it difficult to derive any meaning from the figure. The authors need to improve the resolution and clarity of the figures and verify the pdf generated before final submission.

Decision letter (RSOS-181430.R0)

03-Oct-2018

Dear Miss Jain:

Manuscript ID: RSOS-181430

Title: "Ultra bright emission from Sr doped TiO₂ nanoparticles through r-GO conjugation"

Thank you for submitting the above manuscript to Royal Society Open Science. Your paper was sent to reviewers and their comments are included at the bottom of this letter.

In view of the concerns raised by the reviewers, the manuscript has been rejected in its current form. However, a new manuscript may be submitted which takes into consideration these comments.

Please note that resubmitting your manuscript does not guarantee eventual acceptance, and that your resubmission will be subject to peer review before a decision is made.

Your resubmitted manuscript should be submitted by 02-Apr-2019. If you are unable to submit by this date please contact the Editorial Office.

Yours sincerely,
Dr Laura Smith, MRSC
Publishing Editor, Journals
Royal Society of Chemistry,
Thomas Graham House,
Science Park, Milton Road,
Cambridge, CB4 0WF, UK

Royal Society Open Science - Chemistry Editorial Office

On behalf of the Subject Editor Professor Anthony Stace and the Associate Editor Dr Ya-Wen Wang

REVIEWER(S) REPORTS:

Associate Editor Comments to Author ():

RSC Associate Editor:

Comments to the Author:

(There are no comments.)

RSC Subject Editor:
 Comments to the Author:
 (There are no comments.)

Reviewers' Comments to Author:
 Reviewer: 1

Comments to the Author(s)

Mondal et al have investigated the comparative study of TiO₂, TiO₂: Sr and TiO₂: Sr/r-GO (reduced graphene oxide) nano-composites for their optical and structural properties for photocatalytic and antibacterial applications. The study is interesting and can be accepted in this journal after a careful revision: The comments are as below:

1. The authors have discussed the possible applications in photocatalytic and antibacterial applications but in the introduction the appropriate references are missing (<https://doi.org/10.1116/1.4904503>, <https://doi.org/10.1116/1.4904729>). TiO₂ is being used in various applications so include one paragraph by highlighting the same.
2. Figure 1 is very busy and the vertical lines are not clear so please replace the graph with higher resolution. Figure 2 has also same issue.
3. Author should estimate crystalline size of TiO₂:Sr and its nanocomposite with r-GO. The crystalline size compare with TEM particles size.
4. FTIR spectra of nanocomposite given in figure 2 shows variation in transmittance for TiO₂, TiO₂:Sr, TiO₂:Sr/r-GO. Author should explain variation in intensity of Transition and include shift observed in wavenumber for nanocomposites.
5. In Ellipsometry data, I would suggest author should fit it with suitable equation to explain variation in refractive index (n) and wave vector (k) with wavelength.
6. Author should add some recent references to compare their finding with references.

Reviewer: 2

Comments to the Author(s)

The authors of the manuscript "Ultra bright emission from Sr doped TiO₂ nanoparticles through r-GO conjugation" have discussed the enhancement of defect-related PL emission after doping and rGO conjugation. I do not recommend the acceptance of the manuscript based on the following points.

1. The authors have prepared TiO₂ at a calcination temperature of 400 oC. They got a mixture of three peaks; anatase, rutile and brookite. It is understood that anatase and brookite are the metastable phases and can be formed at low calcination temperature and in a low reaction temperature too. But, the formation of rutile phase needs to be supported with proper evidence. Rutile being the most stable form of TiO₂ are usually formed at high calcination temperature.
2. The line "Sr facilitate the brookite phase might be due to polycrystalline nature of Sr related compounds". It is not clearly presented. What types of compounds ? SrO, SrCO₃ or what ? Similarly, graphene helps in the rutile phase formation. Similarly, Graphene helps in the rutile phase formation. The reason provided as due to "..rutile phase owing to the reduction in crystallite size". How is it so ?
3. The authors have mentioned that the location of Sr²⁺ is not clear. This can be understood since one may need several other techniques to verify this. However, the authors can provide some more insights, such as they should mention the ionic size of the dopant and host lattice ions. TiO₂ has TiO₆ as the basic unit. Sr²⁺ has a sufficiently large ionic radii in 6 coordination which is nearly 132 pm. Ti⁴⁺ in 6 coordination has radii of approx. 74 pm. It is quite evident that it would

be hard for Sr²⁺ to replace Ti⁴⁺. Thus, instead of Sr²⁺ on Ti⁴⁺ it might be on the interstitial site also. Furthermore, a difference in the charge on Sr and Ti might also create oxygen vacancies in the system for charge balance. The authors need to consider these points because finally they have related PL with oxygen defects.

4. There is a difference between band-to-band absorption and band gap absorption ? The authors have claimed 300 nm is the band gap absorption of TiO₂ which correspond to 4.13 eV. However, the band gap as they measured from the Tauc plot gives a band gap of 3.75 eV. Is it the direct or indirect band gap they are calculating ?

5. The authors have drawn a line on the absorption edge on the X-axis with Y=0, and mentioned that to be absorption limit ? What is the meaning of this ? Usually, the line drawn on the absorption edge on the X-axis with Y=0 gives an idea of the band gap of the samples. Are they trying to mean this or what ? If that is the case a matching between Fig. 4 and Fig. 5 might happen if the $(ah\nu)^{1/2}$ is considered. The authors have to be clear on this.

6. They have mentioned that the absorption at 300 nm is due to recombination of careers. What I understand, the photoexcited careers recombine radiatively or nonradiatively giving light or heat. Now, the recombination process should give emission. Although they have mentioned that in the last line of the PL study that PL is obtained by UV excitation at 300 nm, They have mentioned that recombination is the result for the absorption at 300 nm. How is it so ???

7. They have mentioned that the indirect band gap of TiO₂ does not allow a bandgap emission. Thus, the authors need to be more clear now to support that the band gap they have calculated in Fig. 5 is direct or indirect.

8. The authors have mentioned that after Sr addition the PL intensity of TiO₂ is intensified. They surmised that a higher recombination rate is beneficial for high photocatalytic activity. Now, photocatalytic activity requires available electron and holes. If the recombination rate is faster, the electrons and holes will not be available for participation in photocatalysis leading to poor photocatalytic activity. They should be more clear on this line ?

9. Why is the enhancement in PL after rGO addition ? Is it because of the fluorescence contribution from rGO or due to enhancement of oxygen vacancies ?

NOTE: The figures are poorly formatted which makes it difficult to derive any meaning from the figure. The authors need to improve the resolution and clarity of the figures and verify the pdf generated before final submission.

Author's Response to Decision Letter for (RSOS-181430.R0)

See Appendix A.

RSOS-190100.R0

Review form: Reviewer 1

Is the manuscript scientifically sound in its present form?

Yes

Are the interpretations and conclusions justified by the results?

Yes

Is the language acceptable?

Yes

Is it clear how to access all supporting data?

Yes

Do you have any ethical concerns with this paper?

No

Have you any concerns about statistical analyses in this paper?

No

Recommendation?

Accept as is

Comments to the Author(s)

Accept

Review form: Reviewer 2

Is the manuscript scientifically sound in its present form?

No

Are the interpretations and conclusions justified by the results?

Yes

Is the language acceptable?

No

Is it clear how to access all supporting data?

Yes

Do you have any ethical concerns with this paper?

No

Have you any concerns about statistical analyses in this paper?

No

Recommendation?

Accept with minor revision (please list in comments)

Comments to the Author(s)

The authors of the manuscript "Ultra bright emission from Sr doped TiO₂ nanoparticles through r-GO conjugation" have modified the manuscript. There is still one important point which is left uncorrected. This must be corrected before arriving at a final decision.

Comment: The authors have commented "Therefore, it can be predicted that Sr²⁺ ions are not substitute in TiO₂ but strontium replaced Ti⁴⁺ site and replacement of each Ti⁴⁺ ion via a Sr²⁺ ion generated two..." on page 2, lines 54-57.

This line will confuse the reader. Authors should understand that "substitute" and "replace" bear

a similar meaning. They have mentioned that Sr²⁺ is not substituting TiO₂. You should be clear that Sr²⁺ is added to TiO₂ lattice; it cannot replace or substitute TiO₂. The ion it will substitute or replace is Ti⁴⁺. But what I commented that because of large ionic radii of Sr²⁺ it might sit on the surface or move to the interstitial position. The presence of SrCO₃ in XRD directly proves that a large fractions of Sr²⁺ are on the surface which interact with surface bound ions and oxidized to different phases.

Therefore, the authors should add the following sentence-

“Considering the large size of Sr²⁺, we suppose that besides Sr²⁺ at the Ti⁴⁺ sites, a large fraction of ions might stay on the surface or move to the interstitial position.”

Finally, replace the line mentioned in my comment with this one-“Furthermore, the substitution of Ti⁴⁺ on Sr²⁺ will generate two oxygen vacancies for charge neutrality in the lattice.”

Decision letter (RSOS-190100.R0)

06-Feb-2019

Dear Miss jain:

Title: Ultra bright emission from Sr doped TiO₂ nanoparticles through r-GO conjugation
Manuscript ID: RSOS-190100

Thank you for submitting the above manuscript to Royal Society Open Science. On behalf of the Editors and the Royal Society of Chemistry, I am pleased to inform you that your manuscript will be accepted for publication in Royal Society Open Science subject to minor revision in accordance with the referee suggestions. Please find the reviewers' comments at the end of this email.

The reviewers and handling editors have recommended publication, but also suggest some minor revisions to your manuscript. Therefore, I invite you to respond to the comments and revise your manuscript.

Because the schedule for publication is very tight, it is a condition of publication that you submit the revised version of your manuscript before 15-Feb-2019. Please note that the revision deadline will expire at 00.00am on this date. If you do not think you will be able to meet this date please let me know immediately.

Best wishes,

Dr Laura Smith
Publishing Editor, Journals

RSC Associate Editor
Comments to the Author:
(There are no comments.)

Reviewer comments to Author:

Reviewer: 1

Comments to the Author(s)

Accept

Reviewer: 2

Comments to the Author(s)

The authors of the manuscript "Ultra bright emission from Sr doped TiO₂ nanoparticles through r-GO conjugation" have modified the manuscript. There is still one important point which is left uncorrected. This must be corrected before arriving at a final decision.

Comment: The authors have commented "Therefore, it can be predicted that Sr²⁺ ions are not substitute in TiO₂ but strontium replaced Ti⁴⁺ site and replacement of each Ti⁴⁺ ion via a Sr²⁺ ion generated two..." on page 2, lines 54-57.

This line will confuse the reader. Authors should understand that "substitute" and "replace" bear a similar meaning. They have mentioned that Sr²⁺ is not substituting TiO₂. You should be clear that Sr²⁺ is added to TiO₂ lattice; it cannot replace or substitute TiO₂. The ion it will substitute or replace is Ti⁴⁺. But what I commented that because of large ionic radii of Sr²⁺ it might sit on the surface or move to the interstitial position. The presence of SrCO₃ in XRD directly proves that a large fractions of Sr²⁺ are on the surface which interact with surface bound ions and oxidized to different phases.

Therefore, the authors should add the following sentence-

"Considering the large size of Sr²⁺, we suppose that besides Sr²⁺ at the Ti⁴⁺ sites, a large fraction of ions might stay on the surface or move to the interstitial position."

Finally, replace the line mentioned in my comment with this one-"Furthermore, the substitution of Ti⁴⁺ on Sr²⁺ will generate two oxygen vacancies for charge neutrality in the lattice."

Author's Response to Decision Letter for (RSOS-190100.R0)

See Appendix B.

Decision letter (RSOS-190100.R1)

25-Feb-2019

Dear Miss Jain:

Title: Ultra bright emission from Sr doped TiO₂ nanoparticles through r-GO conjugation

Manuscript ID: RSOS-190100.R1

It is a pleasure to accept your manuscript in its current form for publication in Royal Society Open Science. The chemistry content of Royal Society Open Science is published in collaboration with the Royal Society of Chemistry.

RSC Associate Editor
Comments to the Author:
(There are no comments.)

Reviewer(s)' Comments to Author:

Appendix A

Reviewer: 1

Mondal et al have investigated the comparative study of TiO₂, TiO₂: Sr and TiO₂: Sr/r-GO (reduced graphene oxide) nano-composites for their optical and structural properties for photocatalytic and antibacterial applications. The study is interesting and can be accepted in this journal after a careful revision: The comments are as below:

1. The authors have discussed the possible applications in photocatalytic and antibacterial applications but in the introduction the appropriate references are missing (<https://doi.org/10.1116/1.4904503>, <https://doi.org/10.1116/1.4904729>).

TiO₂ is being used in various applications so include one paragraph by highlighting the same.

Reply: Authors are thankful to reviewer for giving good suggestion. It can also be useful in dye sensitized solar cells (DSSCs), photocatalysis, gas sensor, lithium batteries and biosensors. TiO₂ photocatalyst have vast applications in air and water purification as well as in deodorization, sterilization and soil proof. Although, for good photocatalytic activity, the recombination of photo-generated electron hole pairs should be very slow. However, for TiO₂ photocatalyst recombination rate is fast which limits its applications. Again, band gap of TiO₂ falls in UV-range and for a good photocatalyst, its absorption should be in visible region. The same is incorporated in first paragraph, introduction part.

2. Figure 1 is very busy and the vertical lines are not clear so please replace the graph with higher resolution. Figure 2 has also same issue.

Reply: As per the suggestion, we have tried our best to enhance the resolutions of the Figure 1 and 2.

3. Author should estimate crystalline size of $\text{TiO}_2\text{:Sr}$ and its nanocomposite with r-GO. The crystalline size compare with TEM particles size.

Reply: We have taken this comment as the opportunity to enhance the influence of the TEM study and given crystalline size and TEM particles size. We have incorporated the same in page 3 and 4 of the revised manuscript.

4. FTIR spectra of nanocomposite given in figure 3 shows variation in transmittance for TiO_2 , $\text{TiO}_2\text{:Sr}$, $\text{TiO}_2\text{:Sr/r-GO}$. Author should explain variation in intensity of Transition and include shift observed in wavenumber for nanocomposites.

Reply: We are thankful to the reviewer for such constructive comment which will definitely enhance the impact of our manuscript. As suggested we have tried to explain the variation of FTIR intensity and shift with proper reference as follows:

“The intensity of these two vibrational bands is higher in $\text{TiO}_2\text{:Sr/r-GO}$ nanocomposites. Such bands are also seen in TiO_2 and $\text{TiO}_2\text{:Sr}$ which might have attributed to the residual carbon content introduced from some precursors such as polyvinylpyrrolidone. Moreover, the intensity and the position of the vibrational peaks between 1000 and 500 cm^{-1} enhanced and shifted slightly to the lower wavenumber in case of $\text{TiO}_2\text{:Sr/r-GO}$ nanocomposites, which might be due to the interaction between r-GO and

TiO₂ nanoparticles”. These are included and marked in page 4 of the revised manuscript.

5. In Ellipsometry data, I would suggest author should fit it with suitable equation to explain variation in refractive index (n) and wave vector (k) with wavelength.

Reply: According to reviewer suggestion, we have done fitting of ellipsometry data with suitable equation and added fitted parameters in revised manuscript. Fitted figures are included in page number 7 of manuscript.

6. Author should add some recent references to compare their finding with references.

Reply: We are thankful to reviewer for his valuable suggestion. We have added some recent references in appropriate place and mentioned below:

1. Jingsheng Cai, Jiali Shen, Xinnan Zhang, Yun Hau Ng, Jianying Huang, Wenxi Guo, Changjian Lin, and Yuekun Lai, *Small Methods* 2018, 1800184 (1-24).
2. P. Martins, C. Ferreira, A. Silva, B. Magalhaes, M. Alves, L. Pereira, P. Marques, M. Melle-Franco and S. Lanceros-Mendez, *Composites Part B: Engineering*, 2018, **145**, 39–46.
3. S. Abdellatif, P. Sharifi, K. Kirah, R. Ghannam, A.S.G. Khalil, D. Erni and F. Marlow, *Microporous and Mesoporous Materials*, 2018, **264**, 84-91.

Reviewer:

2

The authors of the manuscript “Ultra bright emission from Sr doped TiO₂ nanoparticles through r-GO conjugation” have discussed the enhancement of defect-related PL emission after doping and rGO conjugation. I do not recommend the acceptance of the manuscript based on the following points.

1. The authors have prepared TiO₂ at a calcination temperature of 400 °C. They got a mixture of three peaks; anatase, rutile and brookite. It is understood that anatase and brookite are the metastable phases and can be formed at low calcination temperature and in a low reaction temperature too. But, the formation of rutile phase needs to be supported with proper evidence. Rutile being the most stable form of TiO₂ are usually formed at high calcination temperature.

Reply: We are very much thankful to the reviewer for finding out such important fact. We have done a thorough literature review for answering this and we tried our best to explain our survey in support to our findings which are as follows: "A good number of research directed the formation of mixed phase TiO₂ (anatase and rutile) at low temperature below 450°C, reported so far. For example, Wang et al. produced rutile nanorods via the direct hydrolysis of TiCl₄ ethanolic solution in water at as low as 50 °C. Fischer and co-workers produced a mixture of TiO₂ nanoparticles via low-temperature (130°C) dissolution-precipitation on a microfiltration PES membrane with mixed phase compositions of anatase, brookite, and rutile. Lijuan Bu et al. synthesized rutile TiO₂ nanoparticles via treating anatase TiO₂ with concentrated HNO₃ under the hydrothermal conditions operated at 180°C for 24 h. According to the above discussion, it can be concluded that the formation and the transformation of rutile phase is mainly dependent upon the precursor materials and synthesis conditions." The above justifications are added in the main manuscript in page 2 to 3 with proper references.

2. The line "Sr facilitate the brookite phase might be due to polycrystalline nature of Sr related compounds". It is not clearly presented. What types of compounds ? SrO, SrCO₃ or what ? Similarly, graphene helps in the rutile

phase formation. Similarly, Graphene helps in the rutile phase formation. The reason provided as due to "...". How is it so?

Ans: After incorporating Sr to TiO₂ host, small peaks due to the presence of SrCO₃ are observed in XRD data as seen in the Figures 1(b) and 1(c). The formation of SrCO₃ may be due to the decomposition of SrNO₃ into SrO on the surface of TiO₂, which later converted to SrCO₃ after reacting with CO₂ during the annealing at 400°C. The as formed polycrystalline SrCO₃ might be responsible for the formation of brookite phase (Fig. 1(b)). It is reported that the anatase to rutile phase transformation is extremely a size dependent effect and graphene usually prevents the growth of grains/crystallites. The particle agglomerations get prevented on the surface of graphene. Reportedly, during the formation of graphene nano-composite, graphene is formed *in situ* where it acts as the thin base material for the other coexisting components and keep them in dispersed form [ref. 10.1016/j.jallcom.2015.06.087].

The above text has been added in the revised manuscript page 2-3, along with appropriate reference.

3. The authors have mentioned that the location of Sr²⁺ is not clear. This can be understood since one may need several other techniques to verify this. However, the authors can provide some more insights, such as they should mention the ionic size of the dopant and host lattice ions. TiO₂ has TiO₆ as the basic unit. Sr²⁺ has a sufficiently large ionic radii in 6 coordination which is nearly 132 pm. Ti⁴⁺ in 6 coordination has radii of approx. 74 pm. It is quite evident that it would be hard for Sr²⁺ to replace Ti⁴⁺. Thus, instead of Sr²⁺ on Ti⁴⁺ it might be on the interstitial site also. Furthermore, a difference in the charge on Sr and Ti might also create oxygen vacancies in the system for charge balance. The authors need to consider these points because finally they have related PL with oxygen defects.

Reply: Authors are very much thankful for suggesting such positive approach to enhance the quality of our study. As per the literature review, we think that the replacement of a lower sized Ti^{4+} ion via a larger sized Sr^{2+} generates volume compensating oxygen vacancies, which altered the atomic distances that led to distortion of the structure and favoured the brookite structure to be formed [ref. DOI: 10.1039/c6ra26012h]. It is difficult to substitute smaller radius Ti^{4+} (0.75 Å) via a larger radius Sr^{2+} ion (1.18 Å). Therefore, it can be predicted that Sr^{2+} ions are not substitute in TiO_2 but in our case strontium replaces Ti^{4+} site and replacement of each Ti^{4+} ion via a Sr^{2+} ion generated two oxygen vacancies. The above text has been added in the revised manuscript page 2, along with appropriate reference.

4. There is a difference between band-to-band absorption and band gap absorption ? The authors have claimed 300 nm is the band gap absorption of TiO_2 which correspond to 4.13 eV. However, the band gap as they measured from the Tauc plot gives a band gap of 3.75 eV. Is it the direct or indirect band gap they are calculating ?

Reply: We are very much thankful to the reviewer for his valuable comment. We are very sorry for the misinterpretation. There is no difference between band-to-band absorption and band gap absorption. Generally for bandgap calculation the band edge is considered where the absorption starts to decrease. Here in our case, the absorbance of all the samples decreased in the range of 330 to 400 nm and shift of the absorption edge towards longer wavelengths is observed in TiO_2 : Sr and TiO_2 : Sr/r-GO. We have used Wood and Tauc equation to calculate the band gap from UV-VIS absorption spectra. The equation is estimated as $\alpha h\nu = K(h\nu - E_g)^n$, where α -absorption coefficient, ν -frequency of absorbed photon, h -Planck constant, and E_g -

optical band gap. Exponent n value chosen for type of optical transition $n = \frac{1}{2}$ and 2 for direct allowed and indirect allowed transition respectively. TiO_2 have both direct and indirect band gap but here in our observation good fitting observed for $n=2$ as well as the bandgap value found out after the fitting is for direct allowed transition. Therefore, band gap was calculated for direct allowed transition of TiO_2 . This is mentioned in the revised manuscript. By using the Tauc's plot between photon energy ($h\nu$) (plotted along x-axis) and $(\alpha h\nu)^2$ (plotted along y-axis) and by finding out the point where the tangent drawn to the plot cuts the energy axis, we found out the band gap of TiO_2 as 3.75 eV which is given in Figure 6.

5. The authors have drawn a line on the absorption edge on the X-axis with $Y=0$, and mentioned that to be absorption limit? What is the meaning of this? Usually, the line drawn on the absorption edge on the X-axis with $Y=0$ gives an idea of the band gap of the samples. Are they trying to mean this or what? If that is the case a matching between Fig. 4 and Fig. 5 might happen if the $(\alpha h\nu)^{1/2}$ is considered. The authors have to be clear on this.

Reply: We have drawn on the absorption edge on the X-axis with $Y=0$ and this line referred to absorption limit. Since TiO_2 band gap is generally found in 320-400 nm range and for band gap calculation absorption band edge is taken into consideration. Absorption in this range indicate that if radiation of this wavelength is incident on sample then it will absorb those radiation and after that the absorption value abruptly decreases (around 400 nm) and its value then unchanged for further higher wavelength. So we have mentioned that value absorption limit after which absorption is nearly constant. The band gap is found by using Wood and Tauc equation in figure 6 by plotting graph between $(\alpha h\nu)^2$ and photon energy ($h\nu$).

6. They have mentioned that the absorption at 300 nm is due to recombination of carriers. What I understand, the photoexcited carriers recombine radiatively or nonradiatively giving light or heat. Now, the recombination process should give emission. Although they have mentioned that in the last line of the PL study that PL is obtained by UV excitation at 300 nm, They have mentioned that recombination is the result for the absorption at 300 nm. How is it so ???

Answer: We are very sorry for this big mistake. Yes it is very well understood that recombination of photoexcited carriers gives emission and here in our case we have shown the photoluminescence in visible range i.e. from 575 to 700 nm. According to the detail literature review, we have changed the text in the revised manuscript as follows: "The photoluminescence emission results of the synthesized samples revealed the charge separation efficiency of the prepared materials. All samples showed strong absorbance around 266 nm which has been attributed due to the exciton absorption band. It can be seen that all the samples exhibit intense and broad emission band peaked at around 647nm, and the intensity increased significantly with the addition of strontium and r-GO."

The above text was added on page number 6 in revised manuscript.

7. They have mentioned that the indirect band gap of TiO₂ does not allow a bandgap emission. Thus, the authors need to be more clear now to support that the band gap they have calculated in Fig. 5 is direct or indirect.

Reply: We are thankful to reviewer for bringing our attention on optical band gap. We have calculated the band gap by considering direct allowed optical transition.

8. The authors have mentioned that after Sr addition the PL intensity of TiO₂ is intensified. They surmised that a higher recombination rate is beneficial for high photocatalytic activity. Now, photocatalytic activity requires available electron and holes. If the recombination rate is faster, the electrons and holes will not be available for participation in photocatalysis leading to poor photocatalytic activity. They should be more clear on this line ?

Reply: We are very thankful to the reviewer for this critical comment as well as we are very sorry for our serious mistake. Normally, during photoluminescence, photo-induced electrons are prohibited to form free or binding excitons owing to the binding via the oxygen vacancies and defects. Because of this, the PL emission occurs. Therefore, with the increase in oxygen vacancies or defects, the PL emission intensity also enhances usually. However, in photocatalytic reactions, oxygen vacancies and PL related defects normally capture photo-induced electrons, which inhibits the recombination of such photo-induced electrons and holes benefiting the photocatalytic reactions. Simultaneously, oxygen vacancies are advantageous in promoting the oxygen adsorption. Such adsorbed oxygen later interacts strongly with the photo-induced electrons bound by oxygen vacancies. Hence, oxygen vacancies and defects usually favour the photocatalytic reactions in which oxygen dynamically promotes the oxidation of organic materials.

The above discussion is added in the revised manuscript (page 7) along with proper references.

References

- M.N. Rashed, A.A. El-Amin, International Journal of Physical Sciences, 2 (2007) 73.

- A.L. Linsebigler, G. Lu, J.T. Yates, Chemical Review, 95 (1995) 735.

H. Liu, S. Cheng, M. Wu, H. Wu, J. Zhang, W. Li, C. Cao, Journal of Physical Chemistry A, 104 (2000) 7016.

9. Why is the enhancement in PL after rGO addition? Is it because of the fluorescence contribution from rGO or due to enhancement of oxygen vacancies?

Reply: The PL intensity of TiO₂: Sr/r-GO and TiO₂: Sr are appreciably higher than that in TiO₂. This result is very similar with a reported result where the enhanced intensity is reported for TiO₂/r-GO [ref. Appl. Surf. Sci. 319, 8]. According to this report, TiO₂: Sr/r-GO and TiO₂: Sr can produce more photogenerated electron-hole pairs, which are responsible for the enhanced PL intensity in TiO₂: Sr/r-GO and in TiO₂: Sr than that in TiO₂. During photoluminescence, photo-induced electrons are prohibited to form free or binding excitons owing to the binding via the oxygen vacancies and defects. Because of this, the PL emission occurs. Therefore, with the increase in oxygen vacancies or defects, the PL emission intensity also enhances usually. The above text has been added in the revised manuscript page 3, along with appropriate reference.

NOTE: The figures are poorly formatted which makes it difficult to derive any meaning from the figure. The authors need to improve the resolution and clarity of the figures and verify the pdf generated before final submission.

Reply: We are very much thankful for such positive comment. We have tried our best to improve the picture quality and resolution of the figures and we hope the present version is much more appropriate for its acceptance.

Appendix B

To,
The Editor-in-chief,
Royal Society Open Science

Subject : Answer to Reviewers Comments
Manuscript ID: RSOS-181430

Dear Sir,

Please find the enclosed revised manuscript entitled “**Ultra bright emission from Sr doped TiO₂ nanoparticles through r-GO conjugation**” by Mandal *et al.*, submitted in your esteemed journal “Royal Society Open Science”.

We are very thankful to editor and reviewer for partially accepting our manuscript. We have revised and incorporate valuable suggestions of second reviewer. Thank you for giving us an opportunity to improve our manuscript for better readability.

We hope that the paper is suitable for publication and we look forward to hearing from you in due course.

Thanking you,

Sincerely Yours,

(Dr. Jai Singh)

Department of Physics

Dr. H. S. Gour Central University,

Sagar- 470003, M. P. (India)

Reviewer 2:

The authors of the manuscript "Ultra bright emission from Sr doped TiO₂ nanoparticles through r-GO conjugation" have modified the manuscript. There is still one important point which is left uncorrected. This must be corrected before arriving at a final decision.

Comment: The authors have commented "Therefore, it can be predicted that Sr²⁺ ions are not substitute in TiO₂ but strontium replaced Ti⁴⁺ site and replacement of each Ti⁴⁺ ion via a Sr²⁺ ion generated two..." on page 2, lines 54-57. This line will confuse the reader. Authors should understand that "substitute" and "replace" bear a similar meaning. They have mentioned that Sr²⁺ is not substituting TiO₂. You should be clear that Sr²⁺ is added to TiO₂ lattice; it cannot replace or substitute TiO₂. The ion it will substitute or replace is Ti⁴⁺. But what I commented that because of large ionic radii of Sr²⁺ it might sit on the surface or move to the interstitial position. The presence of SrCO₃ in XRD directly proves that a large fractions of Sr²⁺ are on the surface which interact with surface bound ions and oxidized to different phases. Therefore, the authors should add the following sentence-"Considering the large size of Sr²⁺, we suppose that besides Sr²⁺ at the Ti⁴⁺ sites, a large fraction of ions might stay on the surface or move to the interstitial position." Finally, replace the line mentioned in my comment with this one-"Furthermore, the substitution of Ti⁴⁺ on Sr²⁺ will generate two oxygen vacancies for charge neutrality in the lattice."

Answer: We are very thankful to reviewer for his valuable suggestion. We have made correction in revised manuscript and added modified sentence "Furthermore, the substitution of Ti⁴⁺ on Sr²⁺ will generate two oxygen vacancies for charge neutrality in the lattice. Considering the large size of Sr²⁺, we suppose that besides Sr²⁺ at the Ti⁴⁺ sites, a large fraction of ions might stay on the surface or move to the interstitial position."

These lines added in result and discussion section (second paragraph) page number 2.